# A More Diverse Cervical Microbiome Associates with Better Clinical Outcomes in Patients with Endometriosis: A Pilot Study

**DOI:** 10.3390/biomedicines10010174

**Published:** 2022-01-14

**Authors:** Cherry Yin-Yi Chang, An-Jen Chiang, Ming-Tsung Lai, Man-Ju Yan, Chung-Chen Tseng, Lun-Chien Lo, Lei Wan, Chia-Jung Li, Kuan-Hao Tsui, Chih-Mei Chen, Tritium Hwang, Fuu-Jen Tsai, Jim Jinn-Chyuan Sheu

**Affiliations:** 1Department of Obstetrics and Gynecology, China Medical University Hospital, Taichung 404332, Taiwan; d4752@mail.cmuh.org.tw; 2Department of Medicine, School of Medicine, China Medical University, Taichung 404333, Taiwan; 3Department of Obstetrics and Gynecology, Kaohsiung Veterans General Hospital, Kaohsiung 813414, Taiwan; ajchiang490111@gmail.com (A.-J.C.); nigel6761@gmail.com (C.-J.L.); khtsui@vghks.gov.tw (K.-H.T.); 4Institute of Biomedical Sciences, National Sun Yat-sen University, Kaohsiung 804201, Taiwan; may781111@gmail.com (M.-J.Y.); n1255bbb@gmail.com (C.-C.T.); tritium@mail.nsysu.edu.tw (T.H.); 5Department of Pathology, Taichung Hospital, Ministry of Health and Welfare, Taichung 403301, Taiwan; lukemtlai@gmail.com; 6School of Chinese Medicine, China Medical University, Taichung 404333, Taiwan; lclo@mail.cmu.edu.tw (L.-C.L.); leiwan@mail.cmu.edu.tw (L.W.); 7Department of Chinese Medicine, China Medical University Hospital, Taichung 404332, Taiwan; 8Institute of BioPharmaceutical Sciences, National Sun Yat-sen University, Kaohsiung 804201, Taiwan; 9Human Genetic Center, China Medical University Hospital, Taichung 404332, Taiwan; t6783@mail.cmuh.org.tw; 10Department of Biotechnology, Kaohsiung Medical University, Kaohsiung 807378, Taiwan; 11Institute of Precision Medicine, National Sun Yat-sen University, Kaohsiung 804201, Taiwan

**Keywords:** endometriosis, cervical microbiome, stage, deeply infiltrating endometriosis (DIE), pain, CA125, infertility

## Abstract

Infection-induced chronic inflammation is common in patients with endometriosis. Although microbial communities in the reproductive tracts of patients have been reported, little was known about their dynamic profiles during disease progression and complication development. Microbial communities in cervical mucus were collected by cervical swabs from 10 healthy women and 23 patients, and analyzed by 16S rRNA amplicon sequencing. The abundance, ecological relationships and functional networks of microbiota were characterized according to their prevalence, clinical stages, and clinical features including deeply infiltrating endometriosis (DIE), CA125, pain score and infertility. Cervical microbiome can be altered during endometriosis development and progression with a tendency of increased *Firmicutes* and decreased *Actinobacteria* and *Bacteroidetes*. Distinct from vaginal microbiome, upregulation of *Lactobacillus*, in combination with increased *Streptococcus* and decreased *Dialister*, was frequently associated with advanced endometriosis stages, DIE, higher CA125 levels, severe pain, and infertility. Significantly, reduced richness and diversity of cervical microbiome were detected in patients with more severe clinical symptoms. Clinical treatments against infertility can partially reverse the ecological balance of microbes through remodeling nutrition metabolism and transport and cell-cell/cell-matrix interaction. This study provides a new understanding on endometriosis development and a more diverse cervical microbiome may be beneficial for patients to have better clinical outcomes.

## 1. Introduction

Endometriosis is a common female disease and defined as endometrium and endometrial stroma outside the uterine cavity. The prevalence of endometriosis in women of reproductive age is approximately 5–10% [1]. The common symptoms of endometriosis include dysmenorrhea, pelvic pain, abnormal uterine bleeding or infertility, leading to a major physical and mental burden for women with the disease. Clinically, endometriosis was presumed to be related to estrogen stimulation; the symptoms appear after menarche and vanish after menopause [2,3]. Retrograde menstruation, genetic factors and molecular or immunologic defects were suggested as risk factors to promote the development of endometriosis [4,5,6]. Systemic or local inflammatory changes not only related to the growth of endometrial tissue but also possibly connected with reproductive tract infections [7,8]. In recent years, epidemiologic studies showed the association of endometriosis with pelvic infection and chronic inflammation [9,10]. In addition, patients with endometriosis were also found at higher risk to develop certain types of ovarian cancer [11].

The microbiota, which includes bacteria, virus, fungus and other microorganisms, is important to maintain regulating functions of gut, respiratory tract, and other mucosal microenvironments. Emerging evidence have showed the critical roles of microbiome in training major components of the host’s innate and adaptive immune systems [12,13]. The major impacts of such interactions would be the disruption of regulatory networks on immune responses, leading to pathogenesis of immune-mediated disorders. Since the reproductive tract is a mucosal site opening to exterior environments, impairment of host-microbiome interfaces were frequently linked with inflammatory diseases, infertility and several types of gynecological cancer [14,15,16,17]. Imbalance of reproductive tract microbiota could disturb the immune function and promote the inflammation of reproductive tract which could contribute to the pathogenesis of endometriosis [6,18]. Furthermore, microbiota-derived metabolites can influence the microbiome of different organs. For examples, gut bacteria has been known to show great impacts on central neuron system, hormone system and other extraintestinal organs of the host, such as the reproductive system [19,20]. Dysbiosis in gut tract has been found to significantly increase estrogen levels in the circulation, which can subsequently stimulate growth of ectopic endometriotic cells and promote inflammatory activity [21,22,23].

Several attempts have been made to investigate the microbiota continuum along the female reproductive tract and their impacts on the development of endometriosis [23,24,25]. Some species were frequently linked with endometriosis development, including increased *Streptococcus, Pseudomonas* and *Enterobacteriaceae* with diminished *Atopobium* [26,27,28]. Interestingly, microbial communities differ at different sites of reproductive tract [24,25]. More distinct microbial colonization in patients can be detected in cervical mucus and such differences gradually increased when moving upward along the genital tract to uterus and even fallopian tubes. For examples, *Lactobacillus* dominance in vagina is very common not only in healthy women but also in patients with endometriosis, resulting in less significant differences in community diversity. Recent studies in infertility also considered the utero-microbiome (or placental microbiome) as one of possible causes to determine pregnancy outcome [24,29,30]. However, potential contaminations during sample collection and processing limit the replicability of those low biomass microbiota studies [31,32]. The above findings suggest that cervical microbiome may provide a more reliable and useful bio-signature for detecting common diseases in the upper reproductive tract, e.g., endometriosis.

Although the relationship between the microbiome and endometriosis has been established, most of the studies were focused on the composition and abundance of microorganisms with limited attention on the connections with disease progression and pathophysiological features. In this study, profiles of cervical microbiome from 10 healthy women and 23 endometriosis patients were examined by 16S rRNA amplicon sequencing. The florae associated with clinical stage, deeply infiltrating endometriosis (DIE), pain score, CA125 level, and infertility were identified by different methods for beta-diversity analyses. The KEGG functions altered by long-term drifts of microbial community were also investigated, which may provide another perspective to understand how endometriosis and the associated complications develop and progress.

## 2. Materials and Methods

### 2.1. Study Subjects

We collected consecutive patients who would like to join the study. The cervical swab samples were collected from patients who were diagnosed by laparotomy or laparoscopy examines and pathologically proven as endometriosis at the China Medical University Hospital (CMUH) in Taiwan. For the controls, swab samples were collected from women who received regular physiological checks at the same hospital and were proven to be healthy based on examines conducted. Women in the control group showed an age profile matching with that for the patient group. The endometriosis stage of each patient was classified according to the revised guidelines from the American Society of Reproductive Medicine [33]: stage I, minimal; stage II, mild; stage III, moderate; stage IV, severe. The classification for deep infiltrating endometriosis was based on Enzian score [34]. Disease-related clinical features of diagnosed patients, including pain score, plasma CA125 level, and reproductive ability, were collected from the clinical report.

### 2.2. Microbial DNA Sample Preparation and Quality Check

Microbial DNA was extracted from the cervical swab samples by using Genomic DNA Isolation Kit (FairBiotech Corp., Taoyuan, Taiwan). Four hypervariable regions (V3, V4, V5 and V9) of the 16S ribosomal RNA (rRNA) sequence were amplified by PCR. The detail procedure and sequences of the universal primers can be found in the previous study [35,36]. A total of 10 samples from healthy controls and 23 samples from patients (Appendix A) showed clear amplified DNA fragments and were subjected to library preparation using DNA Library Prep Kit (NEB Inc., Ipswich, MA, USA). Amplicon sequencing was conducted with Illumina paired-end platform.

### 2.3. OTU (Operational Taxonomic Units) Clustering and Species Annotation

The raw data were merged by FLASH [37] and filtered by QIIME [38] to get clean data. The chimeric tags in the clean data were detected and removed using UCHIME to obtain the final effective tags [39]. In order to analyze the diversity of species in each sample, all effective tags were grouped by 97% DNA sequence similarity into Operational Taxonomic Units (OTUs) using the Geengenes databank [40]. OTU taxonomic assignments and annotation were made using RDP classifier based on SILVA Database [41]. The relative species, evenness and abundance distribution in each sample were analyzed by alpha-diversity methods, including Shannon and Chao1. Tree graphs of species annotation for each group were constructed by GraPhlAn [42], and phylogenetic diversity heat trees were generated by Metacoder [43].

### 2.4. Comparison of Community Composition between Groups

To compare microbial communities among groups, beta-diversity was calculated to reflect the dissimilarity between groups, including unweighted unifrac [44] and weighted unifrac distance [45]. The data in this distance matrix were further visualized by t-SNE (t-distributed stochastic neighborhood embedding), a nonlinear dimension reduction method [46], To define potential biomarkers with statistical differences among groups, metagenomic features were detected by linear discriminant analysis (LDA) effect size (LEfSe) [47]. In order to emphasize the difference of the dominant species among three samples in each taxonomic rank, the top 10 species were selected and the ternary plot was drawn based on relative abundance [48].

### 2.5. KEGG Functional Annotation of Cervical Microbiome

To investigate the functional profiles associated with the altered cervical microbiome, OTUs with the same function are collectively referred to Ortholog groups (KO entries) in the KEGG ORTHOLOGY database. Each KO contains multiple gene information and plays a role in one or more pathways. The functions governed by altered cervical microbiome were categorized into three levels of metabolic pathways in the KEGG database (https://www.kegg.jp/kegg/pathway.html; assessed on 13 July 2020).

### 2.6. Statistical Analysis

Statistical methods such as t-test, Wilcox test, and one-way ANOVA (Bartlett’s test or post-test for linear trends) were performed to analyze the significance of difference between/among groups. T-test and Wilcox test were utilized for two group comparison, while Wilcox and one-way ANOVA tests were utilized for more than two groups. To compare populations with two individual parameters, two-way ANOVA (Dunnett’s multiple comparison test) and F-distribution analysis were utilized to determine whether they could be equal or not. The data were presented as mean ± S.D., and a *p* value of less than 0.05 was considered as significant difference. The ecological relationships in each paired group was estimated by Spearman rank correlation analysis. In this study, the absolute magnitude of a correlation coefficient was interpreted as 0.1–0.4: negligible correlation; 0.4–0.7: moderate correlation; 0.7–0.9: strong correlation; 0.9–1.0: very strong correlation.

## 3. Results

### 3.1. OTU (Operational Taxonomic Units) Clustering and Species Annotation of Cervical Microbiome in Healthy Women and Endometriosis Patients at Different Stages

Raw reads of paired-end 16S amplicon sequencing were generated from cervical swabs of 10 healthy women and 23 endometriosis patients (11 at stage I-II and 12 at stage III-IV) (Appendix A). After quality control processing, the final effective tags were collected to perform operational taxonomic unit (OTU) clustering and species annotation. Species composition and abundance among different samples were analyzed and tree graph of species annotation for each group were shown in Figure 1 In our study, the major phyla of bacteria in cervical microbiome of Taiwanese women were *Firmicutes* (68.01 %), *Actinobacteria* (15.77 %), and *Bacteroidetes* (7.06 %) (Figure 1a). Although their abundance was relatively low, the presence of *Proteobacteria* members were detected as one unique population in the cervix of patients (Figure 1b), which are frequently linked with inflammatory conditions in human organs and the associated diseases [49]. We categorized the patients into two groups, namely stage I-II (minimal to mild) and stage III-IV (moderate to severe), in this study. Interestingly, the distributions in phyla suggest the tendency of increased *Firmicutes* with decreased *Actinobacteria* and *Bacteroidetes* during endometriosis progression (Figure 1c). Beta-diversity analyses further confirmed differences in cervical microbiomes between healthy women and endometriosis patients (Figure 1d). In addition, mild difference can also be detected between patients at stage I-II and patients at stage III-IV (Figure 1d), suggesting that the cervical microbiota profiles could be dynamic during disease progression.

### 3.2. Distinct Cervical Microbiomes in Healthy Women and Endometriosis Patients at Different Stages

To further understand the characteristic cervical microbiota that associate with clinical stage, LEfSewas performed to study the phylogenetic trees of dominant microorganisms among different groups. The LDA scores and the associated cladogram reveal that members within the *Coriobacteriia* lineage, e.g., *A. vaginae*, could be considered as the most common microbiota in healthy women (Figure 2a). *L. jensenii* or members in *Corynebacteriales*, *Porphyromonadaceae* and *Ruminococcaceae* were associated with patients at stage I-II. Furthermore, *B. breve* and *Streptococcaceae* members, e.g., *S. agalactiae*, were possible biomarkers for patients at stage III-IV. Ternary plots at genus and species levels were also generated to depict relative occurrence of individual OTUs in three different groups. Our data reveal that *Atopobium* member, e.g., *A. vaginae*, and *Prevotella* members, e.g., *P**. bivia* and *P. amnii*, were dominant in healthy women (Figure 2b), which can be further validated by one-way ANOVA (Figure 2c).

Members from *Streptococcus*, *Bifidobacterium*, *Rhodococcus*, and *Bacteroides* were more related to endometriosis development. Especially, *S. agalactiae* was highly associated with patients at stage III-IV (Figure 2b,c). Distinct from *P. intermedia* and *P. amnii*, *P. bivia* was up-regulated and associated with patients at stage III-IV. Although the abundance of *Lactobacillus* members did not show differences among these three groups, *L. jensenii* was found unique in endometriosis development (Figure 2b,c). *R. erythropolis P. gingivalis* and *B. breve* were found to be associated with patients at stage I-II by ternary plot (Figure 2b). However, the data are not supported by one-way ANOVA. The consistent tendency of *P. gingivalis* along with disease development is not observed, either.

### 3.3. Microbial Flora in Cervial Mucus That Associates with Deeply Infiltrating Endometriosis (DIE)

Deeply infiltrating endometriosis (DIE) is an aggressive type of endometriosis that invades into peritoneal tissues of the pelvic organs, leading to distortion of those organs in incorrect positions. DIE therefore makes surgery complicated and recurrence of DIE remains a major issue in clinical management [50,51]. To know whether microbial flora could be one of risk factors for DIE development, we compared the cervical microbiomes between patients with DIE and patients without (Appendix A). Our data revealed that patients with DIE show increased *Tenericutes* and *Spirochaetes* of the top-10 phyla (Figure 3a) as well as increased *Streptococcus* and *Prevotella* of the top-10 genera (Figure 3b). At the species level, a total of 14 annotated microbes show significant differences in abundance (Figure 3c). The most significant ones include *C. sp.*, *D. micraerophilus*, *F. intestinalis*, *T. berlinense*, *P. intermedia* and *H. macacae*.

### 3.4. Alterations of Cervical Microbiome Correlate with CA125 Level and Pain Score

To confirm the clinical relevance, we further studied the cervical microbiome profiles in patients with different CA125 levels (Figure 4a, left), a plasma biomarker which has been proven to correlate with stage and lesion size of endometriosis. On the other hand, pelvic pain is another common clinical feature for women with endometriosis. However, it does not necessarily correlate with disease stage or CA125 level [52,53], thus the alteration associated with pain score was also performed (Figure 4a, right). It is noteworthy that patients with lower CA125 levels (≤35 U/mL) (Figure 4b, left) or pain scores (≤5) (Figure 4b, right) showed higher OTU enrichments as compared to their counterparts, indicating more diverse cervical microbiotas in patients with mild symptoms. Higher abundances in phyla of *Actinobacteria*, *Tenericutes* and *Chlamydiae*, as well as lower abundance of *Epsilonbacteraeota* were found in patients with higher CA125 levels or pain scores (Appendix A). The abundances of *Bacteroidetes*, *Fusobacteria*, *Spirochaetes* and *Synergistetes* were also altered along with CA125 levels or pain scores but in opposite tendencies (Appendix A).

Through two-group comparison by t-test, the most significant alterations were up-regulation of *Spirochaetes* and down-regulation of *Epsilonbacteraeota*, which were associated with higher pain scores (Appendix A). At the species level, a community with increased *L. jensenii, B. uniformis, B. dorei* and *B. thetaiotaomicron* and decreased *C. hominis* was associated with higher CA125 levels in patients (Figure 4c). To correlate with higher pain scores, increased abundances of *S. anginosus* and *L. salivarius*, as well as decreased *C. ureolyticus* were found as the unique profile (Figure 4d).

### 3.5. Community Diversity of Cervical Microbiome Is Related to Severity of Endometriosis

Although distinct profiles were observed, both CA125 level and pain score showed the correlation with OTU enrichment. We further classified the patients into four groups by combining these two clinical features (CA125/Pain: −/−, −/+, +/−, +/+). Heat tree analysis was performed to depict community diversity in different patient groups. Apparently, more taxonomic classifications can be found in patients with double-negative features. The diversity of genus was going down along with high pain scores or CA125 levels, and the lowest one was defined in patients with double-positive features (Figure 5a). The Shannon plot also indicates the trend of reduced species richness correlated with endometriosis severity (Figure 5b). To know whether those four patient groups show differences in microbial composition, a t-SNE-based classification model was generated as shown in Figure 5c. Consistent with the data presented by the Shannon plot, the compositional distance between patients with single-positive features is very limited, whereas patients between double-positive and double-negative features represent totally distinct microbial composition.

A total of 24 genera (eight down-regulated and 16 up-regulated) show consistently altered trends in terms of OTU abundances among normal control, double-negative and double-positive patients (Figure 5d). Among them, down-regulation in genera of *Dialister*, *Campylobacter*, *Peptoniphilus*, and *Anaerococcus*, as well as up-regulation of *Coprococcus_3*, *Chlamydia* and *Ruminococcus_1* were shown as statistically significant by one-way ANOVA (Appendix A). In addition to 776 shared annotated OTUs, 207 and 284 unique OTUs were exclusively presented in double-negative and double-positive patients (Figure 5e), leading to functional impacts on 11 potent KEGG pathways (Figure 5f). The commonly up-regulated pathways in double-positive patients include signal transduction, two-component system, selenocompound metabolism, secondary bile acid biosynthesis, and ribosome biogenesis in eukaryotes. In contrast, neurodegenerative diseases, shigellosis, pathogenic *Escherichia coli* infection, lysine biosynthesis, Huntington’s disease, and cysteine and methionine metabolism are commonly down-regulated in the most severe patients.

### 3.6. Community Diversity of Cervical Microbiome Is Related to Endometriosis Associated Infertility

Since endometriosis is considered as one major pelvic factor causing female infertility, we further analyzed the differences in cervical microbiomes among patients with fertility, infertility and cured ones after treatments either by surgery or medication. At phylum level, increased *Firmicutes* population was a notable sign for patients with infertility (Figure 6a and Appendix A). At the same time, reduction in populations of *Bacteroidetes*, *Proteobacteria*, *Tenericutes*, and *Epsilonbacteraeota* were also observed. Interestingly, the *Firmicutes*/*Bacteroidetes* (F/B) ratio in cervical microbiome showed a prognostic value to differentiate patients with fertility or infertility (Figure 6b) [54]. In addition, *Firmicutes*/*Actinobacteria* (F/A) and *Firmicutes*/*Proteobacteria* (F/P) ratios were also found as powerful biomarkers for detecting infertility (Appendix A). Notably, cured patients also showed reduced ratios of F/B, F/A or F/P, same as the fertile patients. Those data indicate that alteration of cervical microbiota composition is one of potent factors involved in endometriosis-associated infertility.

OTU enrichment analysis reveals that fertile patients have higher microbiota diversity as compared to infertile patients, and treatments for infertility can improve the microbiota diversity (Figure 6c). Such findings can be further confirmed by Shannon and Chao1 tests that show consistent correlation with reproductive ability (Figure 6d). The t-SNE model can successfully classify patients into three different groups according to their reproductive ability with the cured population between the fertile and infertile patients (Figure 6e). Phylogenetic heat tree indicates increased *Firmicutes* community in infertile patients as compared to fertile patients, especially for *Lactobacillus* (e.g., *L. jensenii*) (Figure 6f and Appendix A). On the other hand, abundant *Bacteroidetes* community is common for fertile patients, including *Prevotella* (e.g., *P. bivia*) and *Bifidobacterium* (e.g., *B. breve*), which could also be detected in cured patients (Figure 6f and Appendix A). Other minor microbes that associate with reproductive ability include *B. fragilis*, *B. coprocola_DSM_17136*, and *B. vesicularis* (Appendix A). Overall, our data conclude that except for bacterial lineages in the *Firmicutes* phylum, the richness, abundance distributions, and diversity of other taxonomic lineages are less in patients with infertility. Such microbial imbalance may serve as a potent driving force to assist the pathogenesis of infertility.

### 3.7. Ecological Dynamics of Cervical Microbiome Associate with Altered Biological Processes in Fertile Patients

To understand the relationship between microbial dynamics and reproductive ability, a heatmap of cervical microbiota composition at genus level was prepared (Figure 7a). With an attention on the recovery of reproductive ability (cured vs. infertile patients), the abundances of genera including *Sneathia*, *Atopobium*, *Dialister*, *Ureaplasma* and *Prevotella_9* were increased after the treatment. On the other hand, the communities of *Gardnerella*, *Lactobacillus*, and *Ruminococcus_2* were reduced when the infertile patients regained their reproductive ability. The long-term drift of microbial community manipulated by infertility treatment was significant, leading to alterations in ecological relationships among cervical microbes (Appendix A). Looking at the correlation matrices for significant changes presented by the three different patient groups, a table showing correlation coefficients between three taxa (*Dialister*, *Bifidobacterium*, and *Prevotella*) and 3 taxa (*Blautia*, *Bacteroides*, and *Faecalibacterium*) was noted that can differentiate infertile patients from fertile or cured patients (Figure 7b).

To further understand how the community drift influences biological processes, the average functional comparisons among these three patient groups were performed according to KEGG categories. A total of 112 KEGG/L3 (from 280) pathways were found which differ significantly between fertile and infertile patients (Figure 8a). Among them, 82 pathways were consistently up- or down-regulated in fertile and cured patients (Figure 8b), and they can be categorized into 8 major KEGG/L2 pathways. The top-3 altered KEGG/L2 functions are amino acid metabolism (66.1 %), cofactors/vitamins metabolism (30.0 %), and transport and catabolism (1.7 %) (Figure 8c). Considering individual variations, one-way ANOVA further indicate nine KEGG/L3 pathways correlated with the community drifts of cervical microbiome (Figure 8d). Those include ascorbate/aldarate metabolism, energy metabolism, isoflavonoid biosynthesis, N-glycan biosynthesis, lipoic acid metabolism, D-arginine/D-ornithine metabolism, arachidonic acid metabolism, pores ion channels, and glycan biosynthesis and metabolism, suggesting the involvement of nutrition metabolism/transportation and cell-cell/cell-matrix interactions in the pathogenesis of endometriosis-associated infertility.

## 4. Discussion

In this pilot study, we investigated the cervical microbiome profiles of endometriosis patients with different clinical features. As compared to normal women, members in the phylum of *Proteobacteria* were predominantly associated with these patients (Figure 1b). Interestingly, increased *Proteobacteria* appears to be associated with endometriosis across various microbiome sites [55,56,57], suggesting the natural property of ineffective immunity and long-term inflammation during endometriosis development. In addition, members in the Phylum of *Firmicutes* were increased in patients as compared to normal women and correlated with advanced stages and infertility (Figure 1c, Figure 5a and Appendix A). With such microbiome profiles, the increased abundance of *Streptococcus* (e.g., *S. agalactiae*) in cervical mucus is common in patients which was also confirmed by a previous study [28], especially for those with DIE (Figure 3b). Consistent with the previous finding [27,58], *Atopobium* members (e.g., *A. vaginae*), a cause to trigger bacterial vaginosis, were suppressed in the cervix of patients (Figure 2c). Although *Lactobacillus* is abundant in cervical mucus, *L. jensenii* colonization was quite unique in patients (Figure 2c) and its abundance correlated with increased CA125 levels (Figure 4d) and infertility (Figure 6f and Appendix A). *L. jensenii* colonization was also reported to be associated with chronic pelvic pain in a study of viginal microbiome [59]. Overall, patients with more severe clinical symptoms including higher CA125 levels, more severe pain and infertility, tend to have cervical microbiomes with lower alpha-diversities, less richness and more imbalanced distribution (Figure 4b, Figure 5b and Figure 6c,d), which can be differentiated by phylogenetic heat tree analyses (Figure 5a and Figure 6f) and t-SNE plots (Figure 5c and Figure 6e). Those data not only indicate the involvement of cervical microbiome in endometriosis development, but also suggest their potent roles in regulating the pathogenesis of the associated complications.

In this study, we noticed the impacts of microbial relationships on clinical features, which could be more meaningful than the abundance of one particular population, especially for those with high inter-individual variations. For examples, the abundance of *Firmicutes* did not show differences in patients with different CA125 levels or pain scores (Appendix A), whereas the community ratios between *Firmicutes* and *Actinobacteria*, (F/A) or *Firmicutes* and *Proteobacteria* (F/*p*) did tell the differences (Appendix A). The other example would be the correlation matrices between three taxa (*Dialister*, *Bifidobacterium*, and *Prevotella*) and three taxa (*Blautia*, *Bacteroides*, and *Faecalibacterium*) which can differentiate infertile patients from fertile or cured patients (Figure 7b), even though those genera, except for *Dialister*, did not show correlation with reproductive ability (Figure 7a). Certain types of symbiotic relationships, including mutualism, amensalism commensalism or parasitism [60], may exist in cervical mucus that determine the pathogenesis of endometriosis-associated complications. Since symbiotic relationships have been linked to metabolic phenotypes [61] and regulation of mucosal immunity [62], more efforts should be made to investigate their roles in the development and progression of endometriosis.

KEGG annotation revealed possible functional pathways governed by the altered cervical microbiome. When comparing the most severe patients (severe pain and high CA125 levels; +/+) with mildest patients (−/−), several upregulated pathways were related to abnormal signal transduction and uncontrolled cell proliferation (Figure 5f). For instance, secondary bile acid synthesis can generate higher reactive oxygen or nitrogen species, leading to genome instability and carcinogenesis [63,64]. Selenocompounds have been considered as potent cancer preventive or anti-cancer agents due to the fact that they influence energy metabolism and cell cycle checkpoints [65]. Enhancement in selenocompound metabolism may subsequently enhance cancer onset and tumor growth. Ribosome biogenesis was recently found upregulated during endometriosis progression and associated with malignant transition [66]. In addition, *Chlamydia* infection, one of the upregulated genera in the (+/+) patients (Figure 5d), has been associated with increased risks to develop cervical or ovarian cancer [67,68]. *Ruminococcus_1*, another upregulated genus (Figure 5d), was also defined as a tumor-associated microbiota [69,70]. Those findings suggest possible links of altered cervical microbiome with more aggressive phenotypes in endometriosis patients. To our knowledge, this is the first evidence to support the existence of cervical microbiome-cancer axis during endometriosis development. However, whether the most severe (+/+) patients with the altered cervical microbiome (Figure 5d) are at a higher risk to develop endometriosis-associated malignancies still requires further investigation.

For patients with infertility, an overall reduction of community richness and diversity was found in cervical mucus (Figure 6c–f and Figure 7a) with increased abundance of members in *Firmicutes* (Figure 6a and Figure 8a). Therefore, not only the well-known increased F/B ratio was detected in infertile patients (Figure 6b and Appendix A), increased F/A and F/P ratios could also contribute to reproductive ability (Appendix A). Interestingly, previous study on the dynamic of gut microbiome during pregnancy found an overall increase of *Proteobacteria* and *Actinobacteria* which can impact host metabolism, providing more energy and benefits for pregnant women [71]. Therefore, F/A and F/P ratios may not only be utilized as potential biomarkers for predicting infertility, but also reflect the requirement of a metabolic network for a successful pregnancy by providing more energy for the host and fetus. KEGG functional prediction also supports this point of view that after the anti-infertility treatment, the major upregulated pathways in treated patients were the ones involved in amino acid metabolism, cofactors/vitamins metabolism, and nutrient transport and catabolism (Figure 8c). How such host-microbial interaction in the reproductive tract, especially in the cervical mucus, can contribute to a better pregnancy outcome needs more detailed investigation.

One of the advantages in our study is that the samples were collected from the cervical canal. Microbiota studies on female reproductive tract have revealed high correlations and consistency between the cervix and peritoneal fluid samples [24,25], indicating the potential of investigating the peritoneal fluid microbiota via the minimally invasive approach of collecting samples from the cervical canal. For example, increased *Proteobacteria* was not only detected in cervical mucus of endometriosis patients in this study, but also found in the peritoneal fluid of patients [57]. The cervix is the portal of vagina connecting to the uterine cavity with the mucus plug functioning as a gatekeeper to prevent infections from ascending from the vagina. Therefore, the microbial community of cervix and endometrium could be different from the vaginal one [24,25]. For examples, *Lactobacillus* produce lactic acid to keep the vagina at low pH values which could benefit women in preventing the growth of pathogenic bacteria. However, such an acidic microenvironment can influence the function of the cervical mucus. This may explain why the colonization of *Lactobacillus*, especially *L. jensenii*, is associated with poor clinical outcomes in patients. Similarly, for certain “bad” microbes in vagina, such as *Atopobium* and *Dialister*, their presence in cervical mucus can be conversely linked with healthy women or milder conditions in patients.

Clearly, our study and previous reports confirm that changes in the flora of the reproductive tract play an important role in the development of endometriosis and the associated complications [6,18,55,56,72,73,74]. Modulation of the microbial flora by using antibiotics or probiotics/prebiotics therefore can be a feasible way to modify the microbial composition and regain the functions of uterine [72,75]. For examples, treating chronic endometritis with antibiotics or uterine lavage has been demonstrated effective to increase conception rates in women of reproductive age [76,77]. On the other hand, because microbes in the gut have huge impacts on female reproductive tract [19,20,21,22,23], diet-induced microbiota remodeling is considered as another possible therapeutic strategy for treating endometriosis. Studies have indicated that women with high intakes of omega-3 unsaturated fatty acids (PUFAs) have a lower risk of developing endometriosis [78,79]; similar dietary benefits have been also shown in mouse studies, resulting in better anti-inflammatory effects and reduced endometriosis formation [80,81]. In a recent clinical study, simultaneous use of antibiotics and probiotics successfully reshaped the flora in the reproductive tract of patients with better pregnancy outcomes [82]. Those findings provide encouraging data that enables us to develop safer and more effective protocols for clinical investigations. However, we need to keep in mind that those treatments may also introduce physiological consequences in other organs in addition to reproductive tract or gut, more research are needed to advance our understanding of the causal relationships and regulatory mechanisms.

Although our study provides interesting results for possible clinical practice, the relatively small sample size is the major limitation of this study. In addition, the hormone status controlled by proliferation phase, ovulation or secretary phase of menstruation could bias our study results. Validation of those findings by further studies can provide more solid conclusion on how cervical microbiome influences the development and progression of endometriosis and the related complications.

## 5. Conclusions

In this study, we have shown the profound alterations of the cervical microbiome that are associated with development of endometriosis and its related complications. The presented results suggest that more diverse cervical microbiomes could benefit patients with better clinical outcomes. Accordingly, the abundance ratios of F/B, F/A and F/P could be useful for identifying patients at more severe conditions. The alterations of cervical microbiome in patients can manipulate certain functional pathways highly relevant to aggressive phenotypes (e.g., cell proliferation and malignant transformation) as well as regulations in nutrition metabolism, transportation and cell-cell/cell-matrix interaction. Treatments on patients have impacts on the ecological relationships in cervical mucus. How to maintain a proper balance of those relationships in a more diverse community should be an important issue for developing new methods/drugs against endometriosis.

## Figures and Tables

**Figure 1 biomedicines-10-00174-f001:**
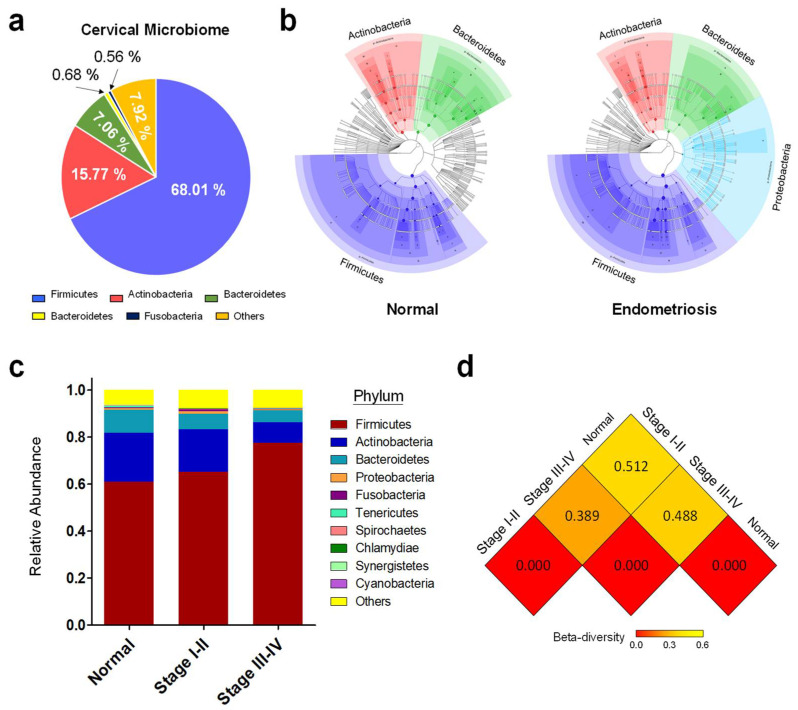
Abundance and composition of cervical microbiomes among normal women and endometriosis patients at different stages. (**a**) The top-5 major phyla of bacteria in cervical microbiome of Taiwanese women. (**b**) Tree graphs of species annotation for cervical microbiomes of normal women and endometriosis patients were constructed and compared by GraPhlAn method. (**c**) Relative abundance of the top-10 major phyla of cervical microbiomes in different groups are shown in distribution histogram. (**d**) Heatmap based on unweighted unifrac distance was plotted to measure the dissimilarity coefficient between pairwise group samples. The three groups include cervical microbiomes from normal women, patients at stage I-II and patients at stage III–IV.

**Figure 2 biomedicines-10-00174-f002:**
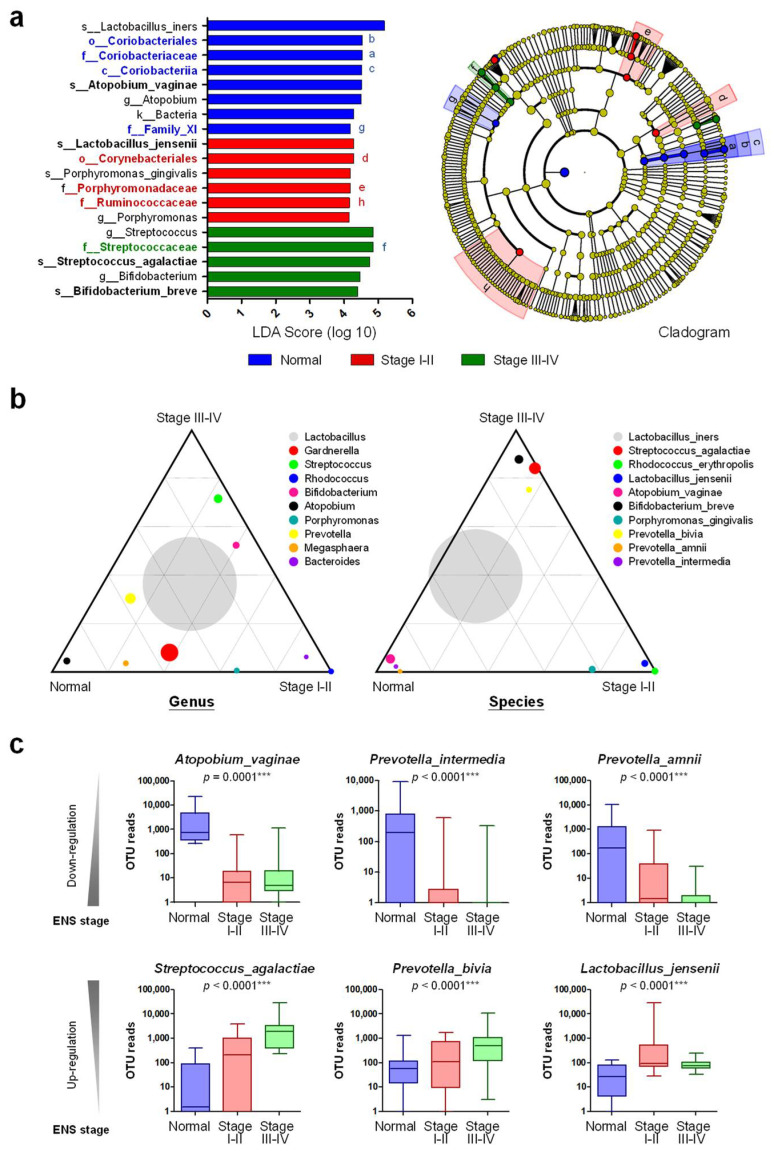
Microbial communities associated with endometriosis patients at different stages. (**a**) LEfSe (linear discriminant analysis (LDA) Effect Size) was performed to define microbial biomarkers for individual groups (normal women, patients at stage I-II, and stage III-IV) with statistical differences. The representative microbiotas for each group are shown with the LDA scores (left panel). The phylogenetic trees of dominant microorganisms are revealed in cladogram (right). (**b**) Ternary plots in Genus (**left**) and Species (**right**) taxonomic ranks were drawn based on relative abundance of top-10 OTUs among different groups. Circles represent dominant species and the size of those circles represent the relative abundance. (**c**) One-way ANOVA was performed to define the major microbes down-regulated (upper) or up-regulated (lower) during endometriosis progression. Statistical significance: ***, *p* < 0.001.

**Figure 3 biomedicines-10-00174-f003:**
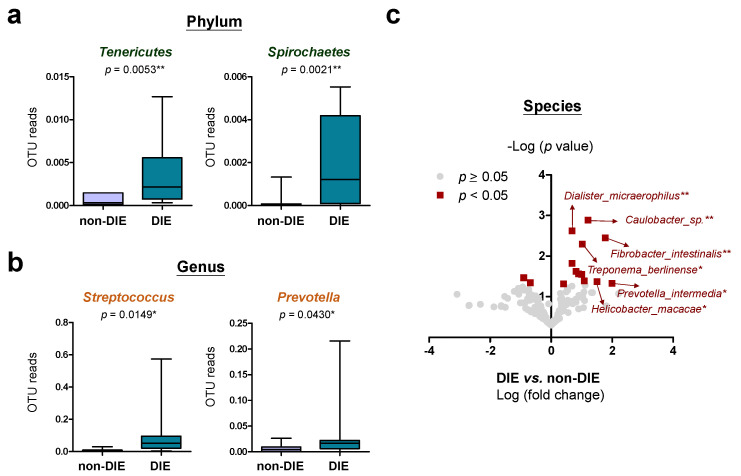
Microbial flora in cervical mucus that associates with deeply infiltrating endometriosis (DIE). Major (**a**) phyla, (**b**) genera and (**c**) species in the cervix that show significant differences in abundance between patients with DIE and patients without were presented. Statistical significance was estimated by t-test and shown as *, *p* < 0.05; **, *p* < 0.01.

**Figure 4 biomedicines-10-00174-f004:**
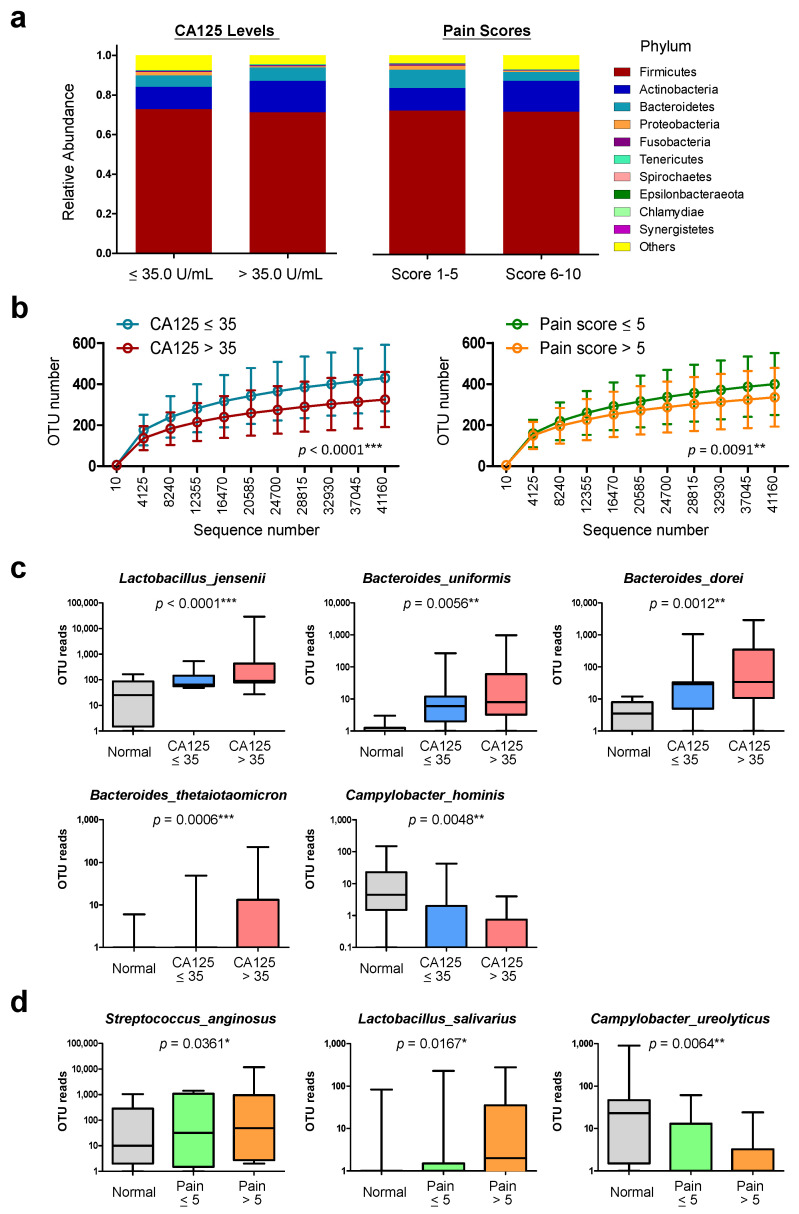
Abundance and composition of cervical microbiomes in endometriosis patients with different CA125 levels and pain scores. (**a**) Relative abundance of the top-10 major phyla of cervical microbiomes in patients with different CA125 levels (**left**) and pain scores (**right**). (**b**) The rarefaction curves indicate the differences in biodiversity (accumulated OTU numbers) of group samples according to CA125 levels (**left**) or pain scores (**right**). Statistics were performed with two-way ANOVA method. One-way ANOVA was performed to define the major down-regulated or up-regulated microbes in cervical mucus of patients that associated with (**c**) higher CA125 levels or (**d**) higher pain scores. Statistical significance: *, *p* < 0.05; **, *p* < 0.01; ***, *p* < 0.001.

**Figure 5 biomedicines-10-00174-f005:**
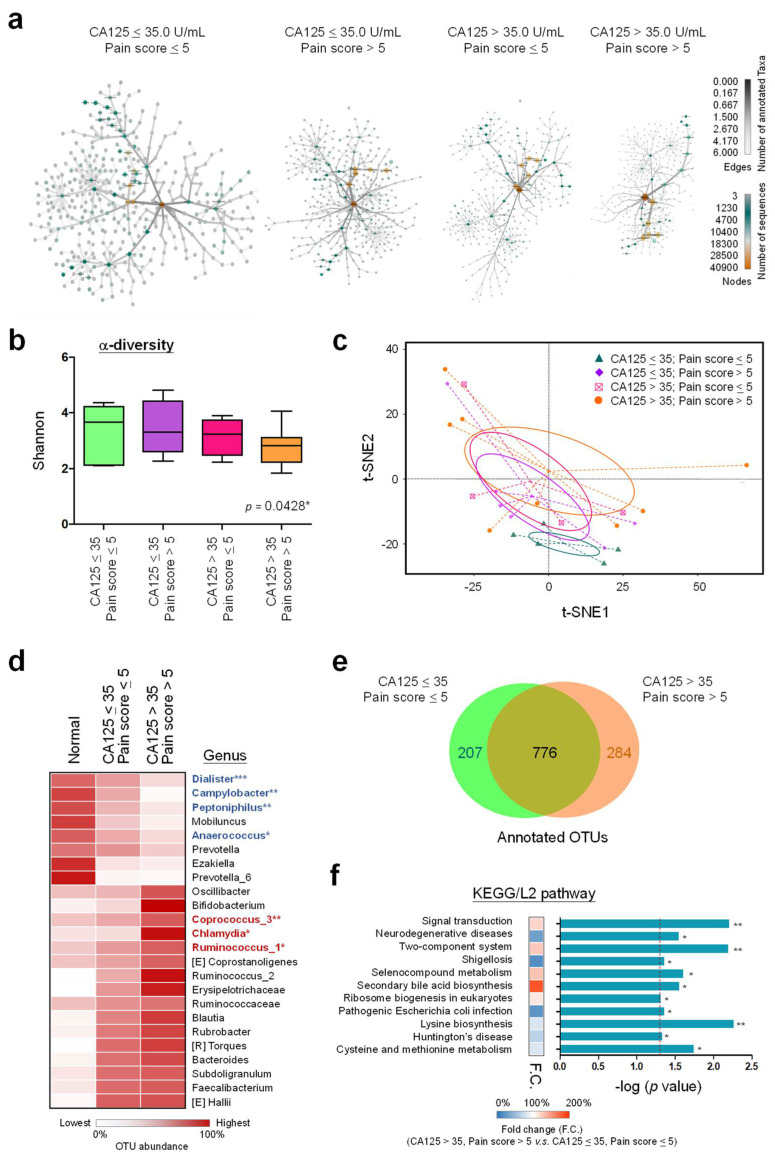
Community diversity of cervical microbiome in endometriosis patients correlate with the severity of clinical outcomes. (**a**) The community diversity in each patient group is revealed by a heat tree map, showing the taxonomic context from higher ranks in the center to lower ranks in peripheries (Kingdom to Species). The abundance of microbiotas in each community can be quantified and visualized by color and size of the nodes and edges. (**b**) The box-plot was generated to illustrate alpha diversity indices (Shannon diversity) in cervical microbiomes of patient groups with different levels of severities. The statistic for the linear trend is estimated by one-way ANOVA. (**c**) The t-SNE analysis diagram was plotted to calculate the actual and embedded distance among the patient groups with different levels of severities. (**d**) The heatmap indicates the average counts of genera that show consistently up-regulated or down-regulated among normal women, double-negative and double-positive patients. Genera with bold font show statistically significant differences in OTU abundance among the groups by one-way ANOVA (Appendix A). (**e**) The Venn diagram indicates annotated OTUs in each group. Values in the overlapping part represent common OTUs. The others are specific OTUs in each group. (**f**) The KEGG/L2 pathways that show significant fold changes between double-positive and double-negative patients are presented. Statistical significance: *, *p* < 0.05; **, *p* < 0.01, ***, *p* < 0.001.

**Figure 6 biomedicines-10-00174-f006:**
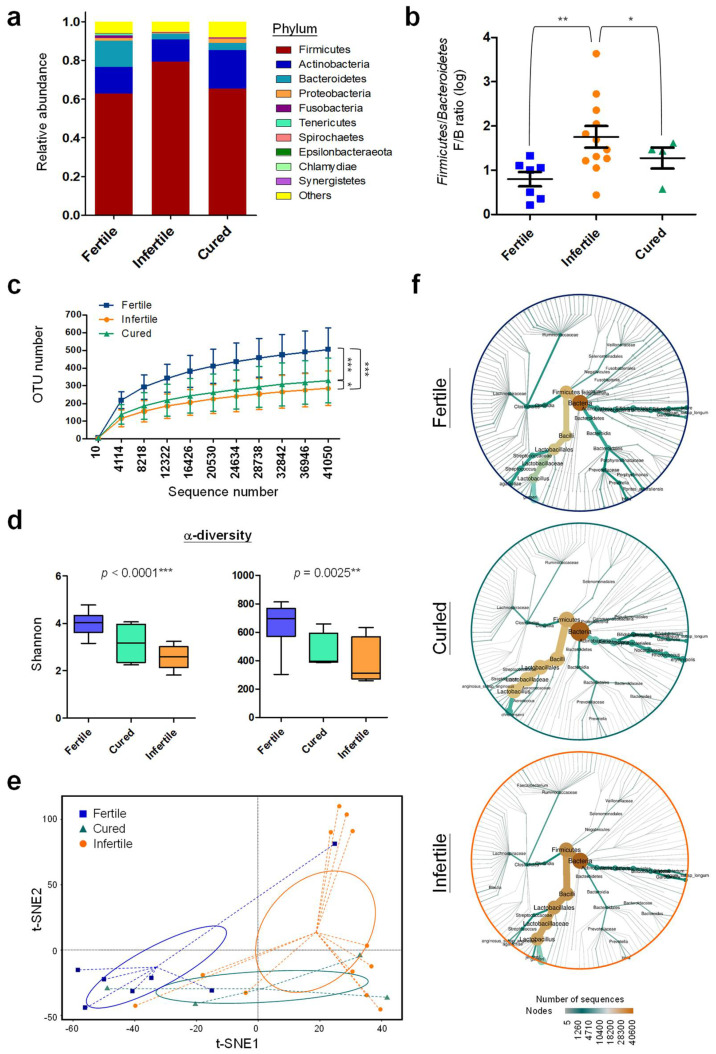
Abundance and composition of cervical microbiomes in endometriosis patients with different reproductive ability. (**a**) Relative abundance of the top-10 major phyla of cervical microbiomes in patients with different reproductive ability are shown in distribution histogram. (**b**) Firmicutes/Bacteroidetes (F/B) ratios in cervical microbiomes of patients between two groups are compared by t-test. (**c**) The rarefaction curves indicate the differences in biodiversity (accumulated OTU numbers) of patients with different reproductive ability. Statistics were performed with two-way ANOVA method. (**d**) One-way ANOVA indicates the trend of alterations in alpha-diversity according to the reproductive ability of patients by using Shannon (**left**) and Chao1 (**right**) methods. (**e**) The t-SNE analysis diagram was plotted to calculate the actual and embedded distance among different patient groups. (**f**) Phylogenetic heat trees were constructed to reveal the abundance and diversity of microbiotas in different patient groups. Statistical significance: *, *p* < 0.05; **, *p* < 0.01; ***, *p* < 0.001.

**Figure 7 biomedicines-10-00174-f007:**
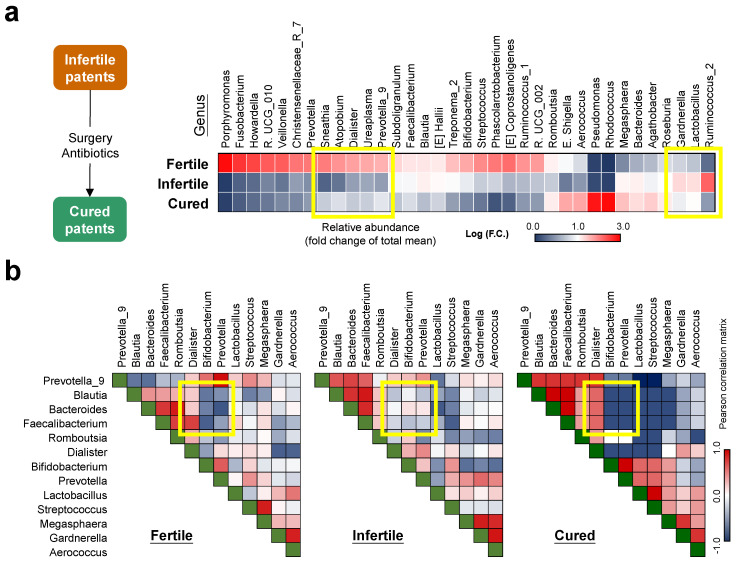
Ecological relationship of bacterial community in cervical mucus in endometriosis patients with different reproductive ability. (**a**) The heatmap reveals relative abundance of genera that show statistical differences between fertile and infertile patients by t-test (*p* < 0.05). Yellow boxes indicate the genera that show consistent correlation between microbial abundance and reproductive ability. (**b**) Spearman correlation matrices reveal inter-genus long-term drifts of cervical microbiomes from patients with different reproductive ability.

**Figure 8 biomedicines-10-00174-f008:**
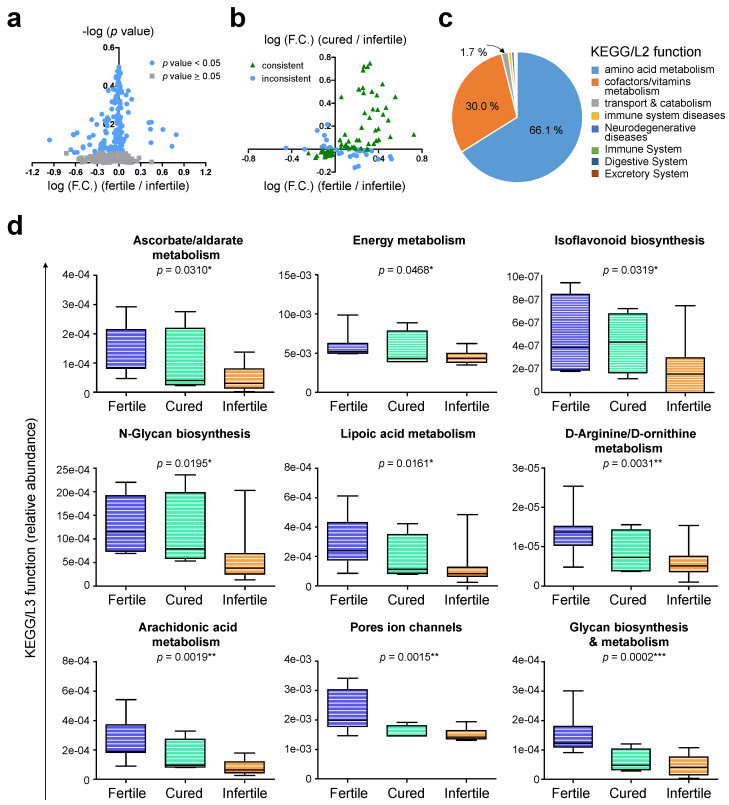
Functional impacts on KEGG pathways governed by altered cervical microbiomes in endometriosis patients with different reproductive ability. (**a**) As compared to fertile patients, the significant altered KEGG/L3 functions (in blue) in infertile patients were plotted by t-test. (**b**) Among the significantly altered KEGG/L3 functions in (**a**), the dot plot indicates the key functions (in green) that can be re-gained or re-suppressed in cured patients by anti-infertility treatments. (**c**) The pie chart indicates the proportions of 8 major KEGG/L2 functions which the key altered functions in (**b**) can be categorized into. The data in (**a**) to (**c**) were generated by using average scores of functional annotations among patient groups. (**d**) Among the key altered functions in (**b**), the box plots indicate nine KEGG/L3 functions which show significant linear trend among patient groups with different reproductive ability by one-way ANOVA. Statistical significance: *, *p* < 0.05; **, *p* < 0.01; ***, *p* < 0.001.

## Data Availability

Data from the experiments presented in this study are included in this published article and its Appendix A. The raw data is available online at the European Nucleotide Archive at the European Bioinformatics Institute, with the accession number PRJEB48793.

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
