# Peer review of "A More Diverse Cervical Microbiome Associates with Better Clinical Outcomes in Patients with Endometriosis: A Pilot Study"

_biomedicines, 2022, doi:10.3390/biomedicines10010174_

Round 1

Reviewer 1 Report

Thank you to giving me the chance to review the paper

despite good overall merit I have some comments:

  • did you perform sample size analysis?
  • did you enroll consecutive patients?
  • Why healty patients were  submitted to Surgery?
  • please differentiate between ovarian and DIE (eg doi: 10.1016/j.jmig.2018.08.031; doi: 10.1080/01443615.2017.1349083)
  • provide some details if available regarding different microbiota composition between the two types

Author Response

despite good overall merit I have some comments,

  1. Did you perform sample size analysis?

Response: This is a pilot study, we didn’t do sample size calculation. The relatively small sample size is one of the limitations of the study. We have added “a pilot study” in title and in Lines 404, 515-516 to address this limitation in Discussion section.

  1. Did you enroll consecutive patients?

Response: We collected consecutive patients who would like to join the study. We have added this sentence in the section of 2.1. Study subjects in Line 105.

  1. Why healthy patients were submitted to Surgery?

Response: Thank you for the correction. The healthy controls did not receive laparotomy or laparoscopy checks. This wrong information has been deleted.

  1. Please differentiate between ovarian and DIE (eg doi: 10.1016/j.jmig.2018.08.031; doi: 10.1080/01443615.2017.1349083) and provide some details if available regarding different microbiota composition between the two types

Response: Thank you for this insightful question. Unfortunately, all the DIE patients we enrolled (Table S1) were diagnosed with having chocolate cysts in the ovary. Therefore, we can only compare the cervical microbiomes between patients with DIE and patients without. Our data revealed that patients with DIE show increased phyla Tenericutes and Spirochaetes as well as increased genera Streptococcus and Prevotella. We have added a new section (3.3. Microbial flora in cervical mucus that associates with deeply infiltrating endometriosis (DIE)) and figure (Fig. 3) to address this issue (Lines 228-239). The original Figs. 3-7 were changed to Figs. 4-8 accordingly. These two suggested references were also inserted into the text, thus the references were renumbered.

Inserted references

  1. Raimondo, D.; Mabrouk, M.; Zannoni, L.; Arena, A.; Zanello, M.; Benfenati, A.; Moro, E.; Paradisi, R.; Seracchioli, R. Severe ureteral endometriosis: frequency and risk factors. J Obstet Gynaecol. 2018, 38, 257-260.
  2. Mabrouk, M.; Raimondo, D.; Altieri, M.; Arena, A.; Del Forno, S.; Moro, E.; Mattioli, G.; Iodice, R.; Seracchioli, R. Surgical, Clinical, and Functional Outcomes in Patients with Rectosigmoid Endometriosis in the Gray Zone: 13-Year Long-Term Follow-up. J Minim Invasive Gynecol. 2019, 26, 1110-1116.

Reviewer 2 Report

The authors compared the cervical microbiome in endometriosis patients with healthy volunteers. The manuscript is well-written, the methods are adequate and the results are well-presented. My minor comments:

  • The main limitation of the study that it is based on the results of relatively small samples (only 23 from patients, and 10 from healthy women). For this reason it should be referred as a pilot study in the discussion section.
  • It is also suggested to compare the results more with the results of other studies since the manuscript contains relatively few references (64).
  • Based on the results more concrete recommendations should be addressed for the future therapy of endometriosis.

Author Response

The authors compared the cervical microbiome in endometriosis patients with healthy volunteers. The manuscript is well-written, the methods are adequate and the results are well-presented.

Response: Thanks for the positive words about our study. We highly appreciate your kind attention and support.

My minor comments:

  1. The main limitation of the study that it is based on the results of relatively small samples (only 23 from patients, and 10 from healthy women). For this reason it should be referred as a pilot study in the discussion section.

Response: Yes, we agree with the reviewer that this study should be considered as a pilot study due to limited sample size. We have added “a pilot study” in title and discussed the limitations which should be further validated by other studies (Lines 515-520) in Discussion section.

  1. It is also suggested to compare the results more with the results of other studies since the manuscript contains relatively few references (64).

Response: Thank you for this important suggestion. We added more discussions to compare our cervical microbiota data with the results reported by other groups. A total of 18 new references were added in the text. Thus, the references were renumbered.

  1. Based on the results more concrete recommendations should be addressed for the future therapy of endometriosis.

Response: Thank you for this constructive suggestion. We have added a new paragraph (Lines 495-514) to discuss the direction for future therapy based on findings from microbiota studies.
